# Histological and Immunohistochemical Analysis of Peri-Implant Soft and Hard Tissues in Patients with Peri-Implantitis

**DOI:** 10.3390/ijerph19148388

**Published:** 2022-07-08

**Authors:** Valentina Flores, Bernardo Venegas, Wendy Donoso, Camilo Ulloa, Alejandra Chaparro, Vanessa Sousa, Víctor Beltrán

**Affiliations:** 1Program of Master in Dental Sciences, Universidad de La Frontera, Temuco 4780000, Chile; v.flores04@ufromail.cl; 2Department of Stomatology, Faculty of Health Sciences, Universidad de Talca, Talca 3460000, Chile; bvenegas@utalca.cl (B.V.); wdonoso@utalca.cl (W.D.); 3Department of Surgical Stomatology, Postgraduate Program in Periodontology, School of Dentistry, Universidad de Concepción, Concepción 4070386, Chile; camulloa@udec.cl; 4Department of Oral Pathology and Conservative Dentistry, Faculty of Dentistry, Universidad de Los Andes, Santiago 7620001, Chile; achaparro@uandes.cl; 5Periodontology and Periodontal Medicine, Center for Host-Microbiome Interactions, Faculty of Dentistry, Oral and Craniofacial Sciences, King’s College London, Guy’s Hospital, London SE1 9RT, UK; vanessa.sousa@kcl.ac.uk; 6Clinical Investigation and Dental Innovation Center (CIDIC), Dental School and Center for Translational Medicine (CEMT-BIOREN), Universidad de La Frontera, Temuco 4780000, Chile

**Keywords:** peri-implantitis, peri-implant disease, APRIL/BAFF, osteonectin, ∝-smooth muscle actin, oral health

## Abstract

Currently, researchers are focused on the study of cytokines as predictive biomarkers of peri-implantitis (PI) in order to obtain an early diagnosis and prognosis, and for treatment of the disease. The aim of the study was to characterize the peri-implant soft and hard tissues in patients with a peri-implantitis diagnosis. A descriptive observational study was conducted. Fifteen soft tissue (ST) samples and six peri-implant bone tissue (BT) samples were obtained from 13 patients who were diagnosed with peri-implantitis. All the samples were processed and embedded in paraffin for histological and immunohistochemical analyses. A descriptive and quantitative analysis of mast cells and osteocytes, A proliferation-inducing ligand (APRIL), B-cell activating factor (BAFF), osteonectin (ON), and ∝-smooth muscle actin (∝-SMA) was performed. We observed the presence of mast cells in peri-implant soft tissue in all samples (mean 9.21 number of mast cells) and osteocytes in peri-implant hard tissue in all samples (mean 37.17 number of osteocytes). The expression of APRIL-ST was 32.17% ± 6.39%, and that of APRIL-BT was 7.09% ± 5.94%. The BAFF-ST expression was 17.26 ± 12.90%, and the BAFF-BT was 12.16% ± 6.30%. The mean percentage of ON was 7.93% ± 3.79%, and ∝-SMA was 1.78% ± 3.79%. It was concluded that the expression of APRIL and BAFF suggests their involvement in the bone resorption observed in peri-implantitis. The lower expression of osteonectin in the peri-implant bone tissue can also be associated with a deficiency in the regulation of bone remodeling and the consequent peri-implant bone loss.

## 1. Introduction

Peri-implantitis (PI) is characterized by the inflammation and destruction of peri-implant bone tissue, leading to implant failure and loss [1]. Currently, the research mainly focuses on inflammatory mediators, cells, or molecules related to the PI etiopathogenesis, which could serve as molecular markers useful in the early diagnosis, prognosis, and treatment of the disease [2,3,4,5,6].

The cytokines A proliferation-inducing ligand (APRIL) and B-cell activating factor (BAFF) are two homologous cytokines of the tumor necrosis factor-α (TNF-α) superfamily. It has been described that APRIL and BAFF control the survival of B lymphocytes in inflammation through the emission of signals, allowing them to complete their maturation process and avoid early elimination [7,8,9]. It has been shown that B cells are the main source of the expression of receptor activator for nuclear factor-κ B ligand (RANKL); therefore, when RANKL binds to its RANK receptor (expressed in osteoclasts), preosteoclast differentiation signals are activated in osteoclasts, promoting bone resorption [10]. In periodontitis, APRIL and BAFF have been studied in gingival crevicular fluid and saliva, where an increase in the expression of these cytokines was observed versus the control group [11,12]. In another study in humans and mice, it was concluded that both BAFF and APRIL support the survival of B cells that express RANKL, thereby contributing to bone loss in chronic periodontitis [13]. However, there are currently no studies that show the expression of APRIL and BAFF in peri-implantitis, and their role in the chronic inflammatory process is unknown.

Osteonectin (ON) is a non-collagenous matrix protein synthesized by osteoblasts, and it plays a critical role in regulating bone remodeling and maintaining bone mass and quality, as it initiates mineralization and promotes mineral crystal formation [14,15]. In peri-implantitis, its role is unknown.

Bone is a specialized mineralized connective tissue, highly vascularized and innervated made up of cells and extracellular matrix. However, the degree of vascularization in the peri-implant bone tissue is unclear. Alpha-smooth muscle actin (alpha-SMA) marks the smooth muscle cells of the blood vessels, as well as the salivary ducts and myoepithelial cells surrounding the salivary gland acini and has been shown to play an important role in fibrogenesis [16,17]. The aim of this study was to analyze and compare the expression of APRIL and BAFF in the peri-implant soft and bone tissue in patients with peri-implantitis; to analyze the expression of osteonectin (ON) and ∝-smooth muscle actin (∝-SMA), to quantify the osteocytes and blood vessels in the peri-implant bone tissue in patients with peri-implantitis, and to quantify the number of mast cells in peri-implant soft tissue in patients with peri-implantitis. Understanding the histological composition of the peri-implant tissues will provide important clues to determine the etiopathogenesis of the peri-implant disease. From this study, the following hypotheses are proposed: there is an expression of APRIL and BAFF in peri-implant soft and bone tissue in patients with peri-implantitis. There is a decreased expression of osteonectin and ∝-SMA in peri-implant bone tissue.

## 2. Materials and Methods

### 2.1. Study Design

A descriptive study was carried out. Fifteen peri-implant soft tissue samples and six peri-implant bone samples were obtained from 13 patients with a diagnosis of peri-implantitis based on the Classification of Peri-implant Diseases and Conditions Workshop 2017 (peri-implant pocket depth more than 5 mm, bleeding on probing with evidence of bone loss) [18]; of them, 5 were males and 8 were females, aged between 33 and 79 years (mean age 60 years). Complete full-mouth periodontal examinations were performed by a periodontist. Four peri-implant pocket depth (PPD) measurements were taken around each implant mesial, distal, buccal, lingual/palatal sites, with a 15-UNC periodontal probe, including bleeding on probing (BOP), suppuration on probing (SOP), peri-implant probing depth, clinical attachment loss (CAL) and plaque index, using a periodontal chart. It was considered BOP and SOP positive (+) when there was bleeding and suppuration at at least one pocket site.

The inclusion criteria considered patients with osseointegrated implants with a follow-up period since their placement equal to or greater than five months, with a diagnosis of peri-implantitis, peri-implant pocket depth more than 5 mm with bone loss greater than 50% and significant cosmetic compromise. The following were excluded from the study: children, adolescents, patients with autoimmune diseases, and pregnant or lactating women. The participants signed an informed consent form at the Faculty of Dentistry, Universidad de La Frontera, Temuco, Chile, and Universidad de Concepción, Chile. The study was conducted in accordance with the principles outlined in the Declaration of Helsinki (2013) on experimentation with human participants. The ethical approval of the study was granted by the Ethics Committee of the Universidad de La Frontera, Report N°024_2018.

### 2.2. Clinical Variables

The variables studied were age (years), sex, location of the implant, smoking habit, previous periodontitis, and systemic chronic diseases (arterial hypertension or diabetes mellitus). Follow-up period since the implants were placed (months), bleeding on probing, suppuration on probing, and type of abutment of the implants were recorded.

### 2.3. Surgical Procedure

Soft tissue biopsies were obtained during the surgical treatment of peri-implantitis. The peri-implant tissue sample was obtained through an internal bezel incision that surrounded the neck of the implant; this tissue was then fixed in formalin and sent for histopathological study. The graft design was performed according to each step to achieve an adequate access and visibility of the area to be treated. Since the removal was not achieved through a flapless approach, it was necessary to perform a flap with an additional vertical releasing incision. The aim was to always prioritize the most conservative approach possible to avoid larger defects in the area. The dimensions of the biopsies were approximately 2 × 3 mm. Each biopsy was immediately placed in 10% buffered formalin for histological analysis.

After that, each implant was removed by a trephine with a diameter sufficient to allow for a margin of at least 0.5 mm between the implant surface and the inner surface of the trephine. The implant was extracted using low speed (50–80 rpm drilling), light pressure and running saline cooling, avoiding clinical sequelae that could alter the subsequent prognosis of a new rehabilitation. The resulting peri-implant bone defect was treated by regenerative bone therapy. All the biopsies were subjected to histopathological analysis at the Oral Pathology Laboratory of the Universidad de Talca.

### 2.4. Histological and Immunohistochemical Processing and Analysis

Tissues were fixed in 10% buffered formalin, sectioned at 5 µm, deparaffinized and hydrated in the typical manner.

Peri-implant soft tissue samples were stained with histochemical and immunohistochemical techniques. Using the Giemsa histochemical technique, mast cells were observed in the connective tissue. The expression of APRIL and BAFF was observed through immunohistochemistry.

In the peri-implant bone tissue samples, a conventional histological technique was used for staining with hematoxylin eosin and an immunohistochemical technique was used for the detection of APRIL and BAFF expression, osteonectin markers, and ∝-smooth muscle actin.

For the immunohistochemical analysis, the sections were incubated with primary antibodies against APRIL (1:250, rabbit polyclonal antibody, Invitrogen, Thermo Fisher Scientific, Waltham, MA, USA), BAFF (1:250, rabbit polyclonal antibody, Invitrogen, Thermo Fisher Scientific, Waltham, MA, USA), osteonectin (1:100, rabbit polyclonal antibody, Invitrogen, Thermo Fisher Scientific, Waltham, MA, USA) and ∝-smooth muscle actin (1:10, mouse monoclonal antibody, Leica Biosystems, Deer Park, Texas, USA). For detection, we used the Vectastain Universal Quick HRP Kit, Peroxidase (Vector Laboratories, Newark, CA, USA) and ImmPACT DAB Substrate, Peroxidase (HRP) (Vector Laboratories, Newark, CA, USA) according to the manufacturer’s instructions.

The reading and interpretation of the results and the capture of images were carried out under optical microscopy (Leica ICC50W, Leica Microscopy Systems, Ltd., Heerbrugg, Switzerland), via the Leica Application Suite, LAS EZ, at 40× magnification. Five microscopic fields were analyzed for each sample. The quantification of the expression of APRIL, BAFF, osteonectin and ∝-actin of smooth muscle was carried out using Image J software (version 1.46j; National Institute of Health, Bethesda, MD, USA) with the application of color deconvolution. The image was decomposed into 3 colors, and a representative color area was selected for automatic quantification (NIH, Bethesda, MD, USA, from https://imagej.nih.gov/ij/docs/examples/stained-sections/index.html accessed on 16 June 2021). The quantification of mast cells, blood vessels, and osteocytes was carried out by analyzing five microscopic fields using Image J software (version 1.46j; National Institute of Health, Bethesda, MD, USA).

### 2.5. Statistical Analysis

The collected data were recorded in a Microsoft Office Excel spreadsheet, in which a descriptive analysis of the data was carried out, determining the mean and its respective standard deviation. The Shapiro–Wilk normality test was performed to evaluate the data distribution. In a normal distribution, a parametric t-test was used for independent samples, and in a no-normal distribution, a non-parametric Mann–Whitney U test was performed. For data analysis, the statistical program IBM SPSS Statistics (version 23.0, Norman Nie, Chicago, IL, USA) was used. A value of *p* ≤ 0.05 was chosen as the significance threshold.

## 3. Results

### 3.1. Clinical Features

The tissue samples were obtained from 13 patients. The mean age was 60 years, with a range from 33 to 79 years, with no significant differences in the mean age between genders (*p* = 0.735). Eight women (61%) and five men (38%) were included. Only two patients were smokers (fewer than 10 cigarettes a day), five patients had chronic diseases such as arterial hypertension (HTN), and only one patient had HTN and diabetes mellitus (DM). The follow-up period since the implants were placed ranged from 5 to 96 months. Regarding the type of abutment, three were with ball attachment and eighteen were cementing. All the implants presented bleeding on probing, and seven presented suppuration on probing. Of the five patients with arterial hypertension, two presented bleeding and suppuration on probing (Table 1 and Table 2).

### 3.2. Histological and Immunohistochemical Results

Cytokine and cell quantification results were divided into soft tissue (PIST) and peri-implant bone tissue with peri-implantitis (PIBT). For each cytokine/protein/cell quantification, mean, standard deviation, minimum and maximum values, were determined (Table 3 and Table 4). The mean percentage of APRIL expression was higher in peri-implant soft tissue (32.17% ± 6.39%) than in peri-implant bone tissue (7.09% ± 5.94%). The mean percentage of BAFF expression was slightly higher in soft tissue (17.26% ± 12.90%) than in peri-implant bone tissue (12.16% ± 6.30%). There was a significant difference between APRIL and BAFF in soft tissues (*p* = 0.001) (Table 5). Regarding the expression of APRIL and BAFF in soft tissues in patients with chronic diseases (HTN/DM), an expression of APRIL ST (34.04) and BAFF ST (24.5) is shown.

In peri-implant bone tissue, the mean of osteonectin percentage was 7.93% ± 3.79%, and approximate ∝-SMA was 1.78% ± 0.718%. Regarding osteonectin, no significant difference was obtained when comparing it with ∝-SMA (*p* = 0.24). The quantification of mast cells in the peri-implant soft tissue resulted in a mean of 9.21 and the osteocyte quantification in bone tissue with a mean of 37.17 (Figure 1 and Table 5).

## 4. Discussion

There is scarce evidence regarding the expression of cytokines in peri-implant soft and hard tissue in humans with peri-implantitis. Some of the studies carried out on peri-implant soft tissues in humans with peri-implantitis have analyzed transforming growth factor beta (TGF-β), interleukin-17 (IL-17), cluster of differentiation 31 (CD31), collagen fibers, and metalloproteinases, among others [19,20,21]. In all studies, the methodology involved biopsies of the interproximal supracrestal peri-implant area or of the site with the deepest probing depth, so it can be deduced that these studies assumed an initial state of peri-implantitis [19,21,22]. Our study was the first to analyze the cytokines APRIL and BAFF in peri-implant bone and soft tissue in humans with peri-implantitis; the biopsies were also the products of explantation in patients diagnosed with peri-implantitis with bone loss greater than 50% and significant aesthetic compromise, so the results could be related to an advanced stage of the disease.

Only one prior study has been carried out in which BAFF and other markers were analyzed in peri-implant crevicular fluid where healthy mucosa, peri-implant mucositis and peri-implantitis were all included. The result of this study showed an increase in BAFF but no significant difference between the groups, suggesting an association with chronic inflammation in peri-implant tissues [23]. Our study provided us with information on the positive expression of these cytokines, supporting the hypothesis that APRIL and BAFF contribute to chronic inflammation in soft tissues and to peri-implant bone tissue resorption. Regarding the expression of APRIL and BAFF in patients with chronic diseases (HTN/DM), it was only appreciated in soft tissues with peri-implantitis and in both cytokines, and it showed a higher expression compared with the total average of the samples.

Regarding osteonectin (ON) in periodontitis, there is still no consensus on its association with alveolar bone loss in said disease [24,25]. Regarding peri-implantitis, a recent study showed no significant differences in ON between the groups (healthy, peri-implant mucositis, and peri-implantitis) in peri-implant crevicular fluid [26]. In our study, the results could be interpreted as a decrease in ON expression resulting in an abnormal bone matrix because of a deficiency in the regulation of peri-implant bone remodeling in advanced stages of the disease.

As to the studies that analyzed vascularization in peri-implantitis, to the best of our knowledge, they have only been performed on peri-implant soft tissue. In our study, vascularization in peri-implant bone tissue with peri-implantitis was analyzed by immunostaining for ∝-SMA, so there are no comparative results under the same conditions. However, it can be compared with studies performed on peri-implant soft tissues. In a similar study, a higher density of blood vessels immune stained with anti-CD31 in peri-implant soft tissue was obtained in the group with peri-implantitis compared to the healthy control, so it is hypothesized that neovascularization contributes to the large influx of leukocytes to the site of inflammation [17]. A recent study obtained similar results. The peri-implantitis group had significantly higher levels of vascular endothelial growth factor (VEGF), CD34, and CD44 expression compared to the other groups (peri-implant health, peri-implant mucositis) [27]. In our study, the blood vessel count and percentage of vascularization through the expression of ∝-SMA were low, which is not surprising, since the greater vascular supply to the peri-implant tissues comes from the supra-alveolar connective tissue or apical compartment to the junctional epithelium in the peri-implant mucosa, and not the peri-implant bone tissue that was the object of study.

In periodontitis, a higher mast cell count has been reported compared to gingivitis [28,29]. In peri-implantitis, a higher mast cell density has been reported, but without significant differences [18]. In our study, a number of mast cells were seen, which could have participated in destructive events or in the defense mechanism against peri-implantitis. However, more studies are needed to elucidate the role of mast cells in peri-implant disease. The main limitations of this study are the number of samples and the lack of a control group, which prevented us from making further inferences. Furthermore, there is a great difference in observation time between the cases (5 to 96 months); therefore, the inflammatory process would be undoubtedly different. Accordingly, for future research, we suggest increasing the number of samples and comparing the expressions of cytokines, proteins, and molecules with a control group to obtain more significant information on the pathogenesis of peri-implantitis and include samples with a similar time period of clinical use. However, the results of this study on patients with an advanced stage of peri-implantitis can be used as a starting point to complement the current understanding of the peri-implant disease.

## 5. Conclusions

There was an expression of APRIL and BAFF in peri-implant bone and soft tissue biopsies with peri-implantitis, which suggested their participation in the chronic inflammation related to the bone resorption of the disease.

There was a deficient expression of osteonectin in peri-implant bone tissue with peri-implantitis, which could be associated with the deficient regulation of bone remodeling and the consequent peri-implant bone loss in advanced stages of the disease.

There were a significant number of mast cells present in the peri-implant soft tissue, which meant that they were actively participating in the inflammatory response.

There was a slight expression of ∝-SMA in bone tissue and a low blood vessel count related to the deficient vascular supply from the peri-implant bone tissue in peri-implantitis.

## Figures and Tables

**Figure 1 ijerph-19-08388-f001:**
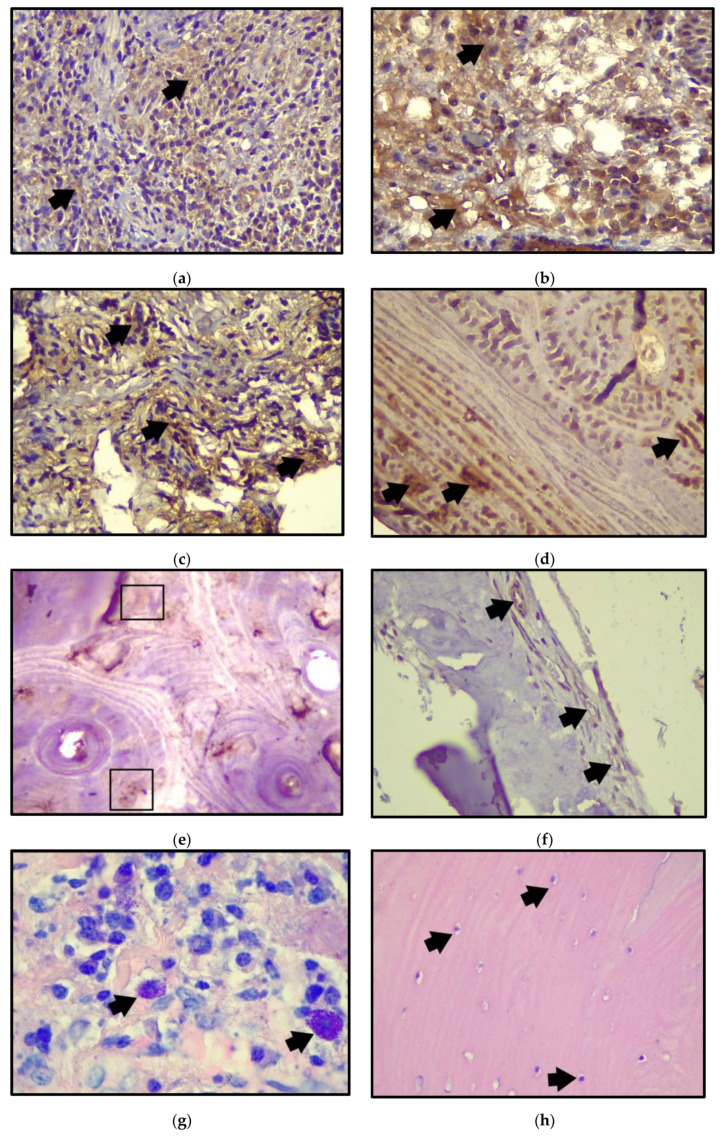
Histological samples of soft and peri-implant bone tissue with peri-implantitis. (**a**) Representative immunohistochemical images for APRIL in peri-implant soft tissue (40×) (black arrows); (**b**) immunohistochemistry for BAFF in peri-implant soft tissue (40×) (black arrows); (**c**) APRIL immunostaining in peri-implant bone tissue with peri-implantitis (40×) (black arrows); (**d**) immunostaining of BAFF in peri-implant bone tissue (40×) (black arrows); (**e**) osteonectin immunostaining in peri-implant bone tissue (40×) (Boxes); (**f**) α-SMA immunostaining in peri-implant bone tissue (40×) (black arrows); (**g**) mast cells in connective tissue in peri-implant soft tissue (Giemsa, 100×) (black arrows); (**h**) osteocytes in peri-implant bone tissue marked in black arrows (40×).

**Table 1 ijerph-19-08388-t001:** Demographic characteristics of patients with peri-implantitis.

	Patients (n = 13)
Sex (M:F)	5:8
Age (mean)	60
Smoking (not/yes)	11:2
Chronic Diseases (absent/present)	8:5
HTN	4
DM	0
HTN/DM	1

M: Male; F: Female; HTN: Arterial Hypertension; DM: Diabetes mellitus.

**Table 2 ijerph-19-08388-t002:** Demographic characteristics of implants with peri-implantitis.

	Implants (n = 21)
BOP (−/+)	0:21
SOP (−/+)	14:7
Follow up (mean)	38.5
Location (mandibular/maxillar)	11:10
Type of abutment (cemented abutment/ball attachment)	17:4

BOP: Bleeding on probing; SOP: Suppuration on probing.

**Table 3 ijerph-19-08388-t003:** Quantification of APRIL, BAFF, mast cells, in peri-implant soft tissue with peri-implantitis.

	APRIL (%)	BAFF (%)	Mast Cells
Mean	32.17	17.26	9.21
Standard deviation	6.39	12.90	2.86
Minimum	23.04	0.34	6
Maximum	47.69	41.89	18

**Table 4 ijerph-19-08388-t004:** Quantification of APRIL, BAFF, osteonectin, α-SMA, blood vessels and osteocytes, in peri-implant bone tissue with peri-implantitis.

	APRIL (%)	BAFF (%)	Osteonectin (%)	α-SMA (%)	Blood Vessels	Osteocytes
Mean	7.09	12.26	7.93	1.78	6.00	37.17
Standard deviation	5.94	6.30	3.79	0.718	2.708	10.420
Minimum	2	4	2	1	2	22
Maximum	17	19	13	2	8	53

**Table 5 ijerph-19-08388-t005:** Relationship of variables APRIL-ST/BT, BAFF-ST/BT, osteonectin, α-SMA.

Variable 1	Variable 2	Value *p*
APRIL-ST ^a^	BAFF-ST ^a^	*p* = 0.001 *
APRIL-BT ^b^	BAFF-BT ^b^	*p* = 0.174
OSTEONECTIN ^c^	α-SMA ^c^	*p* = 0.24

ST, Soft Tissue; BT, Bone Tissue. * There is a statistically significant difference. Shapiro–Wilk normality test, value of *p* ≤ 0.05 was chosen as the significance threshold. ^a^ Student’s *t*-test, *p* < 0.05, ^b^ Student’s *t*-test, *p* > 0.05, ^c^ U-Mann–Whitney test, *p* > 0.05.

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
