# Peer review of "Histological and Immunohistochemical Analysis of Peri-Implant Soft and Hard Tissues in Patients with Peri-Implantitis"

_ijerph, 2022, doi:10.3390/ijerph19148388_

Round 1

Reviewer 1 Report

Introduction: 

  • The role of alpha-SMA should be described.
  • It is necessary to formulate a hypothesis.

MM and results:

  • Number of patients does not match (6 + 5 = 11).
  • Bone loss greater than 50% after five months indicates that the cause of this bone loss is undoubtedly different from the slow inflammatory process. Given the inclusion criterion of 50% bone loss, I suggest not including cases less than 2 years of observation time.
  • A standard description of the probing depth, loss of attachment, and tissue recession among the clinical variables is missing.
  • Data on the type of suprastructure are missing. It was about individual crowns, FPD or completely toothless patients? It is also necessary to define the method of fixing the crowns (cementing/screwing).
  • The surgical technique description lacks a precise description of the incisions or the type of operative process.
  • A description of the bone biopsy and the location of the collection site is missing.
  • Comparing soft tissue and bone samples with statistical tests does not make sense. These are completely different tissues, and the difference is therefore expected, but the fact that it is statistically significant does not provide new information needed to understand the problem of periimplantitis.
  • I suggest a simple description of the results descriptively without comparisons.
  • A comparison with healthy controls would be beneficial.
  • Another option is to compare older and younger cases.

Discussion: 

  • Line 192: Given the nature of the study, it is not possible to talk about causal relations; it should be noted that there is no evidence for these explanations.
  • The same is true for line 203.
  • It should be noted that the absence of alpha-SMA in bone is not surprising.
  • 225: What is a “significant number of mast cells”?
  • Line 234-235: This statement is not accurate because you did not include different stages of periimplantitis.

Correct the discussion according to the new results.

Author Response

  • Comments of the Introduction: 
  • The role of alpha-SMA should be described.

R: The reviewer's suggestion is accepted, so the description of the role of the alpha-SMA is added in the introduction, specifying the importance of studying it. Line 25-30:  “The bone is a specialized mineralized connective tissue, highly vascularized and innervated made up of cells and extracellular matrix. However the degree of vascularization in the peri-implant bone tissue is unclear. Alpha-smooth muscle actin (alpha-SMA) marks the smooth muscle cells of the blood vessels, as well as the salivary ducts and myoepithelial cells surrounding the salivary gland acini and has been shown to plays an important role in fibrogenesis [16, 17].

  • It is necessary to formulate a hypothesis.

R: The reviewer's suggestion is accepted, so the hypothesis is added in the introduction. Line 35-41: From this study, the following hypotheses are proposed: There are alterations in the histological characteristics of human peri-implant soft and bone tissue associated with peri-implantitis compared to those described in the literature on peri-implant health. There is expression of APRIL and BAFF in peri-implant soft and bone tissue in patients with peri-implantitis. There is decreased expression of osteonectin associated with low peri-implant bone quality in the advanced stage of the disease.”

Comments on Materials and Methods, and Results:

  • Number of patients does not match (6 + 5 = 11).

R: The total samples are 21 (15 of the soft tissues and 6 of the hard tissue) of 13 patients. However we corrected a mistake in the description in the sex (5 were men and 8 were women). This information is available in the first line in the material and method section. We appreciate the revisor’s suggestion, however, the sample we chose is very specific, since it requires patients with peri-implantitis with a loss greater than 50%, therefore, it is a difficult sample to obtain. In addition, this study is one of the first studies to study cytokines APRIL/BAFF in peri-implant bone and soft tissue with peri-implantitis. The sample size calculation was complemented with scientific publications with a similar experimental design in which were 18, 14, 15 samples.

 References:

  1. De Araújo, M. F.; Filho, A. F. L.; Da Silva, G. P.; De Melo, M. L. R.; Napimoga, M. H.; Rodrigues, D. B. R.; Alves, P. M.; De Lima Pereira, S. A. Evaluation of Peri-Implant Mucosa: Clinical, Histopathological and Immunological Aspects. Oral Biol. 2014, 59 (5), 470–478. DOI: 10.1016/j.archoralbio.2014.01.011
  2. Borsani, E.; Salgarello, S.; Mensi, M.; Boninsegna, R.; Stacchiotti, A.; Rezzani, R.; Sapelli, P.; Bianchi, R.; Rodella, L. F. Histochemical and Immunohistochemical Evaluation of Gingival Collagen and Metalloproteinases in Peri-Implantitis. Acta Histochem. 2005, 107 (3), 231–240. DOI: 1016/j.acthis.2005.06.002.
  3. Bullon, P.; Fioroni, M.; Goteri, G.; Rubini, C.; Battino, M. Immunohistochemical Analysis of Soft Tissues in Implants with Healthy and Peri-Implantitis Condition, and Aggressive Periodontitis. Oral Implants Res. 2004, 15 (5), 553–559. DOI: 10.1111/j.1600-0501.2004.01072.x
  • Bone loss greater than 50% after five months indicates that the cause of this bone loss is undoubtedly different from the slow inflammatory process. Given the inclusion criterion of 50% bone loss, I suggest not including cases less than 2 years of observation time.

R: We are very grateful for the reviewer’s suggestion. Unfortunately we can’t exclude the cases less than 2 years of observation time, because there would be fewer samples in the study, and the results would lose validity. However, we add this consideration in the discussion, as a study limitation, in line 58 of the Discussion section. “Furthermore, there is a great difference in observation time between the cases (5 to 96 months) therefore the inflammatory process undoubtedly would be different. Accordingly, for future research, we suggest increasing the number of samples and comparing the expressions of cytokines, proteins, and molecules with a control group to obtain more significant information on the pathogenesis of peri-implantitis, and include samples with the time use of clinical similar. However, the results of this study on patients with an advanced stage of peri-implantitis can be used as a starting point to complement to understanding the inflammatory process of peri-implant disease.”

  • A standard description of the probing depth, loss of attachment, and tissue recession among the clinical variables is missing.

R: The reviewer's suggestion is accepted and is added in line 5 of the material and method section: “Complete full-mouth periodontal examinations were performed by a periodontist (V.B), including bleeding on probing, peri-implant probing depth, clinical attachment loss and plaque index, using a periodontogram”.

  • Data on the type of suprastructure are missing. It was about individual crowns, FPD or completely toothless patients? It is also necessary to define the method of fixing the crowns (cementing/screwing).

R: We are very grateful for the reviewer’s suggestion.  We added this data in the results section (line 7) and in the table 2.

  • The surgical technique description lacks a precise description of the incisions or the type of operative process.

R: Following the reviewer’s suggestion, we have added a more detailed description in the 2.3 section of Materials and Methods (lines 109  to 115).

The peri-implant tissue sample was obtained through an internal bezel incision that surrounded all the neck of the implant, this tissue was fixed in formalin and sent for histopathological study.

The graft design was performed according to each step to achieve an adequate access and visibility of the area to be treated. Since the removal wasn’t achieved through flapless, it was necessary to perform a flap with an additional vertical releasing incision. The aim was to always prioritize the most conservative approach possible to avoid larger defects in the area.

  • A description of the bone biopsy and the location of the collection site is missing.

R: Following the reviewer’s suggestion, we have added the description to the 2.3 section of Materials and Methods (lines 118 to 128).

Each implant was removed by a trephine with a diameter sufficient enough to allow for a margin of at least 0.5 mm between the implant surface and the inner surface of the trephine. The implant was extracted using low speed (50-80rpm drilling), light pressure and running saline cooling, avoiding clinical sequelae that could alter the subsequent prognosis of a new rehabilitation. The resulting peri-implant bone defect was treated by regenerative bone therapy.

  • Comparing soft tissue and bone samples with statistical tests does not make sense. These are completely different tissues, and the difference is therefore expected, but the fact that it is statistically significant does not provide new information needed to understand the problem of periimplantitis. I suggest a simple description of the results descriptively without comparisons.

R: We are very grateful for the reviewer’s suggestion.  We eliminated this comparison between soft and bone tissue (Result: line 118).

  • A comparison with healthy controls would be beneficial.

R: We greatly appreciate the reviewer’s suggestion. We are aware the limitations of the study, and they were described in the manuscript (Discussion line: 58). However, the incorporation of control samples to this study presents a bioethical complication, since it would require performing a biopsy of bone tissue around healthy implants, which could only be justified in very specific cases that rarely present in every day clinical work (e.g. damage to the prosthetic connection which is not possible to solve, a poorly positioned implant that cannot be corrected with a soft-tissue approach, among others). Furthermore, this would require a new ethics committee approval and patient consent. However, in relation to the peri-implant soft-tissue biopsies, in future studies we suggest incorporating biopsies from the esthetical gingival remodeling interventions or mucogingival plastic surgery around healthy dental implants that can be used as controls. Besides, when any exceptional situation arises in which it is possible to remove bone tissue around healthy implants, this suggestion will be considered to use these peri-implant bone samples as controls. 

  • Another option is to compare older and younger cases.

R: We greatly appreciate the reviewer’s suggestion. Nevertheless, for this study it would not be possible, because we don’t have enough samples to compare.  However, in future studies this suggestion will be considered.

Comments for the Discussion:

  • Line 192: Given the nature of the study, it is not possible to talk about causal relations; it should be noted that there is no evidence for these explanations.

R: The reviewer's suggestion is accepted, and it was eliminated in the discussion (Line 18 of the Discussion).

  • The same is true for line 203.

R: The reviewer's suggestion is accepted, and it was eliminated in the discussion. (Line 67 of the Discussion).

  • It should be noted that the absence of alpha-SMA in bone is not surprising.

Following the reviewer’s suggestion, a line has been added to the manuscript indicating that “In our study, the blood vessel count and percentage of vascularization through the expression of ∝-SMA were low, which is not surprising, since the greater vascular supply to the peri-implant tissues comes from the supra-alveolar connective tissue or apical compartment to the junctional epithelium in the peri-implant mucosa, and not the peri-implant bone tissue that was the object of study” (line 296 to 300).

  • 225: What is a “significant number of mast cells”?

R: The sentence has been eliminated from the discussion. (Line 51 of the Discussion)

  • Line 234-235: This statement is not accurate because you did not include different stages of periimplantitis.

R: We are very grateful for the reviewer’s suggestion. The reviewer's suggestion is accepted, and it was eliminated in the discussion. (Line 62 of the Discussion).

Reviewer 2 Report

Dear authors, 

The present topic is very interesting and has a high impact for the dental practice.

In section 2.  Materials and Methods

2.1. Study design…….The authors wrote:

A descriptive study was carried out. Fifteen peri-implant soft tissue samples and six peri-implant bone samples were obtained from 13 patients with a diagnosis of peri-implantitis based on the Classification of Peri-implant Diseases and Conditions ….

It is mandatory to be detailed the Classification of Peri-implant Diseases and Conditions.

In section 2.2. Clinical variables ….. The authors wrote:

The variables studied were the age of the patient, sex, location of the implant, smoking habit, and history of diseases (arterial hypertension or diabetes mellitus). The clinical use period of the implant in months and the presence of bleeding and suppuration were recorded.

All those variables must be encountered in the results.

3. Results

3.1. Clinical features

The samples were obtained from 13 patients. The mean age was 60 years, with a range from 33 to 79 years, with no significant differences in the mean age between genders (p = 0.735). Eight women (61%) and five men (38%) were included. Only two patients were smokers (fewer than 10 cigarettes a day), five patients had chronic diseases such as arterial hypertension (HTN), and only one patient had HTN and diabetes melli- tus (DM). The time of clinical use of the implant ranged from 5 to 96 months. All the implants presented bleeding on probing, and only seven presented suppuration.

The demographic data should be presented as a chart (e.g. Patient’s repartition upon age and sex)

Also, must be mentioned the location of the implant within the patients

What were the obtained data in patients with peri-implantitis and chronic diseases such as arterial hypertension (HTN) and diabetes mellitus (DM) ?

  1. Discussion

Must be detailed the expression of cytokines in peri-implant soft and hard tissue in humans with peri-implantitis and chronic diseases such as arterial hypertension (HTN) and diabetes mellitus (DM).

Author Response

Comments of the Materials and Methods

2.1. Study design the reviewer wrote: It is mandatory to be detailed the Classification of Peri-implant Diseases and Conditions.

R: We are very grateful for the reviewer’s suggestion, and it was added in the Materials and Methods section (Line 4 and 11 of the material and methods) “A descriptive study was carried out. Fifteen peri-implant soft tissue samples and six peri-implant bone samples were obtained from 13 patients with a diagnosis of peri-implantitis based on the Classification of Peri-implant Diseases and Conditions Workshop 2017 (peri-implant pocket depth more than 5 mm, bleeding on probing with evidence bone loss)”

2.2. Clinical variables the reviewer refer this phrase: “The variables studied were the age of the patient, sex, location of the implant, smoking habit, and history of diseases (arterial hypertension or diabetes mellitus). The clinical use period of the implant in months and the presence of bleeding and suppuration were recorded” and wrote that: All those variables must be encountered in the results.
R: We greatly appreciate the reviewer’s suggestion. The clinical variables are described in the manuscript. However, we added this data in a table format to facilitate the comprehension (Table 1 and 2).

Comments of the Results

3.1. Clinical features the reviewer suggest that the demographic data should be presented as a chart (e.g. Patient’s repartition upon age and sex). Also, must be mentioned the location of the implant within the patients.

R: We greatly appreciate the reviewer’s suggestion. It was added in the result section. A table with the demographic characteristic clinical of patients and implant has also been added (Table 1 and 2).

What were the obtained data in patients with peri-implantitis and chronic diseases such as arterial hypertension (HTN) and diabetes mellitus (DM) ?

R: We greatly appreciate the reviewer’s suggestion. We reply the reviewer’s questions in the result section: “Of the five patients with arterial hypertension, two presented bleeding and suppuration on probing. However, it is not possible to do more inference regard this data.” (Line 9 of the Result).

 Comments of the Discussion

Must be detailed the expression of cytokines in peri-implant soft and hard tissue in humans with peri-implantitis and chronic diseases such as arterial hypertension (HTN) and diabetes mellitus (DM).

R: We greatly appreciate the reviewer’s suggestion. Regarding the expression of APRIL and BAFF in soft tissues in patients with chronic diseases (HTN/DM), it showed a higher expression over samples total average. This information has been added in the discussion section (Line:  21 of the Discussion). 

Reviewer 3 Report

Dear Authors, I found your article very interesting. As we know peri-implant inflammation is a condition mediated by many factors and it is important to study and learn about them in the best possible way. I find the content of your article very timely and very well done.
However, some considerations must be made:

1)The topic you deal with is very broad, it would be necessary to add some references in the introduction to make it longer.

2How was the sample size calculated?

Author Response

Comments on the Introduction.

  • The topic you deal with is very broad, it would be necessary to add some references in the introduction to make it longer.

R: We greatly appreciate the reviewer’s suggestion.  We added more references in the introduction (Line 71 of the Introduction).

References:

  1. Brennan, P.; Umar T.; Zaki, GA.; Langdon, JD.; Spedding, A.; Buckley, J.; Downie, P. Are myoepithelial cells responsible for the widespread expression of inducible nitric oxide synthase in pleomorphic adenoma? An immunohistochemical study. J Oral Pathol Med. 2000, 29:279-83. DOI: 10.1034/j.1600-0714.2000.290607.x
  2. Kawasaki, Y.; Imaizumi, T.; Matsuura, H.; Ohara, S.; Takano, K.; Suyama, K.; Hashimoto, K.; Nozawa, R.; Suzuki, H.; Hosoya, M. Renal expression of alpha-smooth muscle actin and c-Met in children with Henoch-Schonlein purpura nephritis. Pediatr Nephrol. 2008, 23(6):913-9. DOI: 10.1007/s00467-008-0749-6
  • How was the sample size calculated?

R: The sample size calculation was performed based on the approximate prevalence of peri-implantitis at the Implantology Polyclinic of The Universidad de La Frontera, which is 15%, and complemented with scientific publications with a similar experimental design in which there were 18, 14, 15 samples.

References:

  1. De Araújo, M. F.; Filho, A. F. L.; Da Silva, G. P.; De Melo, M. L. R.; Napimoga, M. H.; Rodrigues, D. B. R.; Alves, P. M.; De Lima Pereira, S. A. Evaluation of Peri-Implant Mucosa: Clinical, Histopathological and Immunological Aspects. Oral Biol. 2014, 59 (5), 470–478. DOI: 10.1016/j.archoralbio.2014.01.011
  2. Borsani, E.; Salgarello, S.; Mensi, M.; Boninsegna, R.; Stacchiotti, A.; Rezzani, R.; Sapelli, P.; Bianchi, R.; Rodella, L. F. Histochemical and Immunohistochemical Evaluation of Gingival Collagen and Metalloproteinases in Peri-Implantitis. Acta Histochem. 2005, 107 (3), 231–240. DOI: 1016/j.acthis.2005.06.002.
  3. Bullon, P.; Fioroni, M.; Goteri, G.; Rubini, C.; Battino, M. Immunohistochemical Analysis of Soft Tissues in Implants with Healthy and Peri-Implantitis Condition, and Aggressive Periodontitis. Oral Implants Res. 2004, 15 (5), 553–559. DOI: 10.1111/j.1600-0501.2004.01072.x

Round 2

Reviewer 1 Report

Thank you for responding my queries. I have no additional comments. 

Congratulations for a nice research!

Author Response

Thank you very much for your valuable comments and suggestions on our manuscript.

Reviewer 2 Report

Dear  authors,

I  saw that you have answered to all the aspects that I have noticed.

I consider that the article can be published in these latest version.

Author Response

Thank you very much for your positive evaluation. The comments and suggestions were valuable and very helpful for improving our manuscript